# Contribution of the tobamovirus resistance gene *Tm-1* to control of tomato brown rugose fruit virus (ToBRFV) resistance in tomato

**Avner Zinger[1,2], Adi Doron-Faigenboim[1], Dana Gelbart[1], Ilan Levin[1], Moshe Lapidot [1]***

**1** Department of Vegetable and Field Crop Research, Institute of Plant Sciences, Agricultural Research Organization – Volcani Institute, Rishon LeZion, Israel, **2** The Robert H. Smith Faculty of Agriculture, Food and Environment, The Hebrew University of Jerusalem, Rehovot, Israel

\* lapidotm@volcani.agri.gov.il

## Abstract

Tomato brown rugose fruit virus (ToBRFV) is a rapidly spreading pathogen that poses a significant threat to tomato production worldwide. We previously identified a locus on tomato chromosome 11 controlling tolerance to the virus. We further established that combining this locus with one that maps to the *Tm-1* region on chromosome 2 confers resistance to the virus. Here we sought to determine whether, and how, the *Tm-1* gene itself is involved in ToBRFV resistance. Overexpression of *Tm-1* in a tolerant genotype significantly reduced viral accumulation, conferring resistance to ToBRFV. On the other hand, overexpression of *Tm-1* in a susceptible genotype only delayed symptom appearance. Moreover, effective RNAi-silencing of *Tm-1* in the resistant genotype yielded susceptible plants. These findings show that the *Tm-1* gene interacts genetically with the locus controlling tolerance on chromosome 11 and that this interaction is critical for achieving effective resistance to ToBRFV. In addition, the symptomatic plants obtained following silencing of *Tm-1* in the resistant genotype indicate that tolerance is also dependent on normal expression levels of the recessive *tm-1* allele.

## Author summary

Viruses are a significant threat to agriculture, particularly in crops like tomatoes, causing both yield and quality losses. One such virus, Tomato brown rugose fruit virus (ToBRFV), has been spreading rapidly and severely impacting global tomato production. In our study, we focused on understanding the genetic basis of resistance to ToBRFV. Previously, we identified a genetic locus on tomato chromosome 11 that controls tolerance to the virus. We also discovered that combining this tolerance locus with another located near the *Tm-1* gene on chromosome 2 confers full resistance to the virus. Our research further explored

**Data availability statement:** Raw sequence reads generated using PacBio sequencing and used in this study have been deposited in the NCBI (National Center for Biomedical Information) BioProject database under the accession number PRJNA1193317.

**Funding:** This work was supported by the United States–Israel Binational Agricultural Research and Development (BARD) (award no. IS-5276-20 to ML) and the Chief Scientist of the Israeli Ministry of Agriculture and Rural Development (award no. 20-01-0178 to ML and IL). The funders had no role in study design, data collection and analysis, decision to publish, or preparation of the manuscript.

**Competing interests:** The authors have declared that no competing interests exist.

the role of the *Tm-1* gene itself. We found that overexpressing *Tm-1* in a tolerant genotype significantly reduced viral accumulation, providing resistance. However, when *Tm-1* was overexpressed in a susceptible genotype, it only delayed symptoms. Additionally, silencing *Tm-1* in the resistant genotype rendered the plants susceptible to the virus. These findings show that *Tm-1* interacts with the tolerance locus on chromosome 11, and this interaction is essential for effective resistance to ToBRFV. Moreover, we hypothesize that tolerance is dependent on the normal expression of the recessive *tm-1* allele.

## Introduction

Tomato (*Solanum lycopersicum L.*) is one of the world's most important crops, serving as a primary food source in human nutrition. Tomatoes are rich in essential metabolites and antioxidants, with related health benefits [1]. Tomato crops' adaptability to various climates, including tropical, cool, arid, and desert regions, has contributed to its widespread cultivation. According to the Food and Agriculture Organization of the United Nations (FAO), in 2022, nearly 186 million tons of tomatoes were harvested worldwide, covering approximately 5 million hectares (https://www.fao.org/faostat/en/#data/QCL, accessed March 6, 2025).

Plant viruses pose a serious threat to agriculture, reducing both crop yield and quality, with devastating economic consequences. Among the most destructive are the tobamoviruses, which profoundly impact key crops such as tomato, leading to significant challenges for growers.

Viruses in the genus *Tobamovirus*, family *Virgaviridae*, are characterized by a positive-strand RNA genome that is approximately 6.4 kb in length, and rod-shaped virions [2]. These viruses are classified according to various criteria: host range, serological properties of their viral particles, amino acid composition of their capsid protein, and nucleotide sequence of their genomes. The tobamovirus genome harbors four open reading frames: two replication proteins (RdRp) of 126 kDa and their read-through version (183 kDa), a movement protein (MP) of 30 kDa, and a coat protein (CP) of 17.5 kDa [2,3]. The genus comprises about 35 species [4], with two notable members being tobacco mosaic virus (TMV) and tomato mosaic virus (ToMV) [5]. In contrast to vector-transmitted viruses, tobamoviruses infect plants through microscopic wounds in the tissue and spread via physical contact, which increases their potential for disease transmission. Furthermore, these viruses can persist on seeds and soil, requiring sterilization to ensure virus-free cultivation [5].

Tobamovirus research has been a focal point in agricultural science. Preventive strategies, such as soil and equipment sterilization, crop rotation, and using clean propagation materials (seeds, cuttings, or tissue culture), have been employed to limit the spread of these viruses [5]. While helpful, these methods do not offer long-term protection. Genetic resistance has proven to be a more sustainable approach to reducing crop losses [6], offering relatively efficient management of viral diseases, and providing crop protection without additional labor or material expenses during the growing season [7,8].

Over the past few decades, substantial progress has been achieved in understanding plant-virus interactions. Nearly half of the identified virus-resistance genes in plants exhibit dominant control [9], while the others display a recessive mode of inheritance, often arising from the absence of crucial host factors necessary for viral replication [7]. However, continuous evolution of plant viruses calls for new robust and long-lasting resistance strategies [7].

Advances in breeding have made it possible to enhance genetic resistance to tobamoviruses, particularly ToMV, in cultivated tomatoes. Two essential resistance genes have been identified. One of these, *Tomato mosaic virus resistance 1* (*Tm-1*), was derived from *Solanum habrochaites*. It is semi-dominant and located on chromosome 2. *Tm-1* encodes a ca. 80-kDa protein that binds to and inhibits the replication protein of ToMV [3,10,11]. The second gene, *Tm-2*, originated from *Solanum peruvianum*; it is located on chromosome 9, and has two alleles: *Tm-2* and *Tm-2$^2$* [10,12]. Both alleles show dominant inheritance and encode proteins belonging to the CC-NBS-LRR class of resistance proteins [13]. These proteins disrupt the functionality of the ToMV MP, with *Tm-2$^2$* being more sustainable and thus widely used in tomato breeding programs [8,14]. In 2015, a new tobamovirus, tomato brown rugose fruit virus (ToBRFV), was identified infecting a commercial tomato hybrid, cv. *Candela*, in a greenhouse in Jordan. This virus was associated with mild leaf symptoms and distinctive brown rugose symptoms on the fruit [15]. ToBRFV was also detected in a greenhouse in southern Israel, infecting commercial tomato hybrids carrying the *Tm-2$^2$* resistance gene, indicating the ineffectiveness of this gene in controlling ToBRFV [16,17]. It was shown that changes in the ToBRFV MP sequence, compared to that of ToMV, allow it to evade *Tm-2$^2$* while maintaining its ability to spread [18]. The impact of ToBRFV was severe, causing dense mosaic patterns on leaves, narrowed foliage, and yellow-spotted fruit, and resulting in significant losses in tomato quantity and quality [19]. Moreover, ToBRFV can invade the reproductive tissues of tomatoes, including pollen grains; however, although it negatively impacts pollen germination, it is not transmitted through pollen [20]. Levitzky et al. [21] discovered that pollinating bumblebees (*Bombus terrestris*) can mechanically disseminate ToBRFV through buzz pollination, demonstrating its strong transmission potential. Since its initial identification, ToBRFV has spread globally, affecting major tomato-producing countries such as China, India, Turkey, Italy, Spain, and the United States [22–26], among other countries (https://gd.eppo.int/taxon/TOBRFV/distribution, accessed March 6, 2025)

Since its appearance, efforts have been invested in identifying genetic sources of resistance to ToBRFV. These efforts have resulted in several patents claiming the discovery of resistant or tolerant sources on chromosomes 2, 6, or 9 of the *Solanum* genome (https://patentscope.wipo.int/search/en/result.jsf?_vid=P10-LX1RC5-87423, accessed March 6, 2025). In addition, 4 out of 44 tomato accessions screened by Kabas et al. [27] showed tolerance to ToBRFV. Furthermore, Jewehan et al. [28] screened 636 *Solanum* accessions and identified 3 with high resistance to ToBRFV and 26 tolerant accessions. These authors further assessed wild tomato accessions (*S. habrochaites* and *S. peruvianum*) for their response to infection with ToBRFV. Among the 173 accessions studied, nine *S. habrochaites* accessions and one *S. peruvianum* accession demonstrated significant resistance [29]. These plants remained asymptomatic at 24°C, with no detectable virus in the inoculated leaves. However, upon exposure to 33°C, leaves of the inoculated plants developed mosaic patterns and deformations, indicating susceptibility to the virus at higher temperatures. This research group also reported the discovery of a newly identified mutant isolate of ToBRFV capable of overcoming the previously documented resistance [30]. Another study, by Jaiswal et al. [31] evaluated 476 accessions from 12 different *Solanum* species, and identified 44 accessions with resistance or tolerance to ToBRFV.

We previously identified a single ToBRFV-resistant genotype (VC554) and 29 ToBRFV-tolerant ones [32]. Among the latter, VC532 was selected for a detailed study, along with VC554. We found that the tolerance trait is controlled by a single recessive gene located on chromosome 11. In contrast, the resistance trait identified in VC554 was controlled by two genes: one located on chromosome 11, allelic to the gene controlling tolerance in VC532, and another locus that mapped to the *Tm-1* region on chromosome 2. However, *Tm-1* alone did not affect ToBRFV infection [32]. It therefore remained unclear whether the *Tm-1* gene itself participates in this resistance.

PLOS Genetics

The main objective of the present study was to further study the role of *Tm-1* in conferring resistance to ToBRFV. Based on our previous association studies and considering the established role of *Tm-1* in controlling resistance to ToMV [3], we hypothesized that *Tm-1* itself participates in conferring resistance to ToBRFV, potentially through genetic interaction with the locus controlling tolerance on chromosome 11.

We employed two approaches to demonstrate the direct involvement of *Tm-1* in regulating the resistance trait. In the first, we downregulated the expression level of *Tm-1* in the resistant genotype VC554 using RNA interference (RNAi). The second approach involved overexpression of the *Tm-1* gene in the tolerant VC532 and susceptible 'Moneymaker' genotypes. In addition, to study the architecture of the *Tm-1* locus and validate non-synonymous nucleotide changes within the *Tm-1* coding sequence (CDS), we used fourth-generation sequencing PacBio technology to sequence the genomes of the three main genotypes participating in this study: the susceptible 'Moneymaker' line, the tolerant VC532 line, and the resistant VC554 line.

## Results

### The *Tm-1* locus is associated with ToBRFV resistance

In our previous report [32], we showed that combining the *Tm-1* resistance allele with the locus controlling tolerance on chromosome 11, originating from either the tolerant genotype VC532 or the resistant genotype VC554, yields ToBRFV-resistant plants. On the other hand, combining the susceptible allele *tm-1* with those genotypes yielded tolerant plants. To confirm that *Tm-1*, in combination with the locus controlling tolerance on chromosome 11, is truly associated with ToBRFV resistance, we inoculated 60 $F_3$ seedlings segregating at the *Tm-1* locus while fixed for the tolerance QTL. Virus-accumulation analyses, individual plant genotyping, and ToBRFV symptom assessment using a disease severity index (DSI) (Fig 1), were conducted 30 days post-inoculation (DPI). Results presented in Tables 1 and S1 show that all of the tested plants displayed a symptomless phenotype (DSI = 0), as expected due to the effect of the QTL on chromosome 11. Nonetheless, only plants carrying *Tm-1* (homozygous *Tm-1/Tm-1* and heterozygous *Tm-1/tm-1)* exhibited a significant reduction in virus accumulation [$P(F) = 2.54 \times 10^{-15}$, $R^2 = 0.69$], compared to plants that were homozygous for the susceptible allele (*tm-1/tm-1*). These results confirmed that *Tm-1* is strongly associated with the resistance phenotype in plants carrying the tolerance locus in a homozygous state. However, it remains uncertain whether *Tm-1* itself, or a gene or genes that are closely linked to it, contributes to the resistance. To study the direct involvement of *Tm-1* in the resistance phenotype, we modulated the expression of the gene using two approaches: RNAi silencing and overexpression.

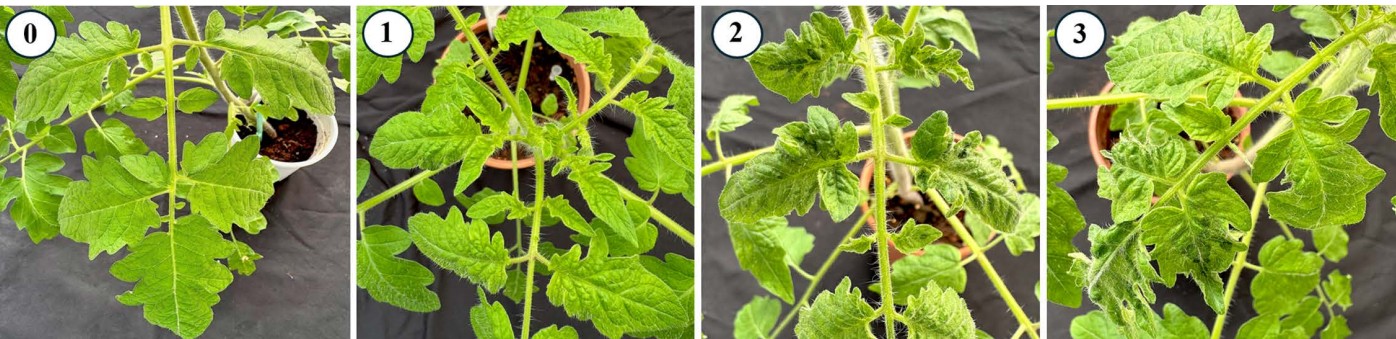

**Fig 1. Tomato brown rugose fruit virus (ToBRFV) disease severity index (DSI).** ToBRFV-induced symptoms on tomato leaves were evaluated according to a DSI consisting of: (0) no visible symptoms; (1) mild mosaic pattern observed on the apical leaf; (2) distinct mosaic pattern on apical leaves; (3) pronounced mosaic pattern accompanied by notable elongation or deformation of apical leaves.

**Table 1. _Tm-1_ is associated with the resistance phenotype in plants carrying the tolerance locus.**

| | Genotype | N | DSI | ToBRFV level (OD,405 nm) | ToBRFV-level range (OD,405 nm) |
|---|---|---|---|---|---|
| **Control lines** | Moneymaker | 4 | 3[A] | 355[B] ± 66 | 204-420 |
| | VC532 | 4 | 0[B] | 834[A] ± 62 | 661-938 |
| | VC554 | 4 | 0[B] | 0[C] ± 0 | 0-0 |
| **F₃ plants** | _11^VC532_/_11^VC532_ _tm-1/tm-1_ | 16 | 0[B] | 457[B] ± 40 | 248-749 |
| | _11^VC532_/_11^VC532_ _Tm-1/tm-1_ | 29 | 0[B] | 16[C] ± 13 | 0-388 |
| | _11^VC532_/_11^VC532_ _Tm-1/Tm-1_ | 15 | 0[B] | 0[C] ± 0 | 0-0 |

The table presents average disease severity index (DSI), average ToBRFV level and range in ToBRFV-inoculated control lines and $F_3$ plants segregating for _Tm-1_ at 30 days post inoculation (DPI). Results are presented as mean ± standard error; N denotes the number of plants tested; average viral level and viral-level range are presented as optical density (OD) × 1000. $11^{VC532}$ is the locus controlling tolerance on chromosome 11; 'Moneymaker' represents a susceptible genotype, VC532 is the tolerant genotype, and VC554 is the resistant genotype. The segregating $F_3$ plants originated from an initial cross between VC532 and VC554. Means with different superscript letters are significantly different at $P < 0.05$, based on Tukey–Kramer HSD test.

## Overexpression of _Tm-1_ in the tolerant VC532 genotype

To further validate the interaction between _Tm-1_ and the locus controlling tolerance on chromosome 11, we overexpressed _Tm-1_ in the tolerant genotype VC532. This line is homozygous for the locus controlling tolerance on chromosome 11. Of 35 transgenic $T_0$ plants generated, 11 were randomly selected for _Tm-1_ expression analysis. Expression levels of _Tm-1_ in these plants were 4–1000 times higher than in the non-transgenic VC532 (S2 Table). $T_0$ plants were self-pollinated to produce $T_1$ populations segregating for the _Tm-1_ overexpression transgene (_Tm-1^OE_). Three $T_1$ populations: TM-127, TM-130, and TM-134, were randomly selected for further analysis and to generate $T_2$ populations homozygous for the transgene.

Analysis of the $T_2$ plants involved three pairs of populations, each consisting of transgenic and non-transgenic azygous control plants originating from each of three independent $T_1$ populations. The results of ToBRFV inoculation and _Tm-1_ expression analysis for the $T_2$ populations and the control genotypes are summarized in Tables 2 and S3.

Following ToBRFV inoculation, neither of the two control genotypes carrying the locus controlling tolerance on chromosome 11 (VC532 and VC554) showed disease symptoms (DSI = 0). In contrast, the susceptible 'Moneymaker' genotype displayed severe disease symptoms at 30 DPI. However, ToBRFV-accumulation levels were high in both the tolerant and susceptible genotypes. As expected, only the resistant genotype VC554, harboring both the locus controlling tolerance on chromosome 11 and the ToMV-resistant gene _Tm-1_, had very low virus levels (Table 2A).

None of the $T_2$ plants displayed disease symptoms following ToBRFV inoculation (DSI = 0) due to their genetic tolerance background conferred by the locus on chromosome 11. However, a consistent pattern was observed in each of the three $T_2$ transgenic populations, consisting of elevated _Tm-1_ expression coupled with a highly significant reduction in virus-accumulation levels compared to their azygous control plants (Table 2B). This reduction in virus levels rendered the transgenic plants resistant to the virus and strongly suggests that _Tm-1_ controls resistance in combination with the locus controlling tolerance on chromosome 11.

## Overexpression of _Tm-1_ in a susceptible genotype

Our previous findings indicated that the _Tm-1_ gene alone is ineffective at controlling either tolerance or resistance to ToBRFV [32]. We therefore overexpressed the gene in susceptible 'Moneymaker' plants to validate these results.

**Table 2. Overexpression of *Tm-1* in the tolerant genotype VC532.**

|  | Line | Description | Type | BR | DSI | ToBRFV level (OD, 405 nm) | Fold *Tm-1* transcription |
|---|---|---|---|---|---|---|---|
| **A.** | **Control genotypes** | | | | | | |
|  | Moneymaker | $11^{MM}/11^{MM},tm\text{-}1/tm\text{-}1$ | Non-transgenic control | 3 | $2.6^{A}$ | $471.4^{AB} \pm 117$ | 1 |
|  | VC532 | $11^{VC532}/11^{VC532},tm\text{-}1/tm\text{-}1$ | Non-transgenic control | 3 | $0^{B}$ | $617.7^{A} \pm 148$ | 0.3 |
|  | VC554 | $11^{VC554}/11^{VC554},Tm\text{-}1/Tm\text{-}1$ | Non-transgenic control | 3 | $0^{B}$ | $14.3^{B} \pm 14$ | 0.02 |
| **B.** | **Original T₁ parental lines** | | | | | | |
|  | TM-127 | $11^{VC532}/11^{VC532},Tm\text{-}1^{OE}/Tm\text{-}1^{OE}$ | T₂ transgenic | 3 | $0^{A}$ | $0^{B} \pm 0$ | 13.3 |
|  |  | $11^{VC532}/11^{VC532},tm\text{-}1/tm\text{-}1$ | T₂ azygous control | 3 | $0^{A}$ | $526^{A} \pm 52$ | 1 |
|  | TM-130 | $11^{VC532}/11^{VC532},Tm\text{-}1^{OE}/Tm\text{-}1^{OE}$ | T₂ transgenic | 3 | $0^{A}$ | $1^{B} \pm 1$ | 57.4 |
|  |  | $11^{VC532}/11^{VC532},tm\text{-}1/tm\text{-}1$ | T₂ azygous control | 3 | $0^{A}$ | $476^{A} \pm 45$ | 1 |
|  | TM-134 | $11^{VC532}/11^{VC532},Tm\text{-}1^{OE}/Tm\text{-}1^{OE}$ | T₂ transgenic | 3 | $0^{A}$ | $24^{B} \pm 21$ | 29.8 |
|  |  | $11^{VC532}/11^{VC532},tm\text{-}1/tm\text{-}1$ | T₂ azygous control | 3 | $0^{A}$ | $506^{A} \pm 29$ | 1 |

The table presents average disease severity index (DSI), average ToBRFV level, fold *Tm-1* transcription in control plants (A) and in tolerant VC532 T₂ plants overexpressing *Tm-1* compared to their azygous control counterparts (B) at 30 days post inoculation (DPI). Results are presented as mean ± standard error; 11 represents the genetic source of the locus controlling tolerance on chromosome 11, where $^{MM}$ stands for the susceptible 'Moneymaker' allele, while $^{VC532}$ and $^{VC554}$ signify the allele controlling tolerance originating from the tolerant and resistant genotype, respectively; $Tm\text{-}1^{OE}$ symbolizes the introduced *Tm-1* overexpression transgene; BR denotes the number of biological repeats, each composed of five different T₂ or parental plants. Average virus levels are presented as optical density (OD) × 1000. *Tm-1* transcription is presented as range and fold change compared to the 'Moneymaker' genotype (A) and compared to the azygous control genotype of each pair (B). Means with different superscript letters are significantly different at $P < 0.05$, based on Tukey–Kramer HSD test (A) or pair-wise comparisons using Student's *t*-test (B).

Introducing *Tm-1* under the control of the *35S* promoter into 'Moneymaker' plants yielded 17 T₀ plants that displayed a diverse range of *Tm-1* expression levels, from 0.15 to 18 times higher than those observed in non-transgenic 'Moneymaker' plants (S4 Table). T₀ plants were self-pollinated to produce T₁ populations segregating for $Tm\text{-}1^{OE}$. We chose the first three T₁ populations (TM-184, TM-185, and TM-186) to produce seeds and generated three T₂ populations homozygous for the transgene and three azygous counterpart populations. The three pairs of homozygous T₂ populations, alongside their respective control genotypes, were inoculated with ToBRFV and analyzed at 20, 30, and 46 DPI (Tables 3, and S5-S7).

*Tm-1* expression level in the T₂ transgenic plants was, on average, 3- to 14-fold higher than in their non-transgenic azygous counterparts. However, despite this elevated expression, virus-accumulation levels in the transgenic plants were not significantly different from their non-transgenic azygous counterparts. Moreover, they did not differ significantly from the virus level in the control genotypes. Symptom severity, on the other hand, was significantly and consistently lower in the transgenic plants compared to their azygous control counterparts and, at most time points, compared to the control 'Moneymaker' and LA2825 (*Tm-1*/*Tm-1* genotype in cv. Moneymaker) lines. The relatively higher symptom levels characterizing the transgenic T₂ plants originating from the TM-186 T₁ line can be attributed to their lower-fold average increase in *Tm-1* expression levels compared to the other two transgenic lines. Together, these results indicated that overexpression of *Tm-1* in a susceptible genotype can yield plants that are at best tolerant to ToBRFV, and they substantiate the hypothesis that a genetic interaction between *Tm-1* and the tolerance locus on chromosome 11 is required to yield high ToBRFV resistance.

## Silencing *Tm-1* expression in the resistant VC554 genotype

To further understand the role of *Tm-1* in ToBRFV resistance, we introduced an RNAi silencing vector (*Tm-1^{AS}*) into the resistant VC554. The five resultant T₀ transformants displayed a broad spectrum of *Tm-1* expression suppression, with

**Table 3. Overexpression of *Tm-1* in the susceptible genotype.**

| | Line | Description | Type | BR | DSI 20 DPI | DSI 30 DPI | DSI 46 DPI | ToBRFV level (OD, 405 nm) | Fold *Tm-1* transcription |
|---|---|---|---|---|---|---|---|---|---|
| **A.** | **Control genotypes** | | | | | | | | |
| | Moneymaker | $11^{MM}/11^{MM}, tm\text{-}1/tm\text{-}1$ | Non-transgenic control | 3 | $2.7^A$ | $3.0^A$ | $2.4^A$ | $756.8^A \pm 143.6$ | 1 |
| | LA2825 | $11^{LA2825}/11^{LA2825}, Tm\text{-}1/Tm\text{-}1$ | Non-transgenic control | 3 | $1.2^B$ | $2.0^A$ | $2.5^A$ | $452.5^A \pm 78.0$ | 3 |
| **B.** | **Original T$_1$ parental lines** | | | | | | | | |
| | TM-184 | $11^{MM}/11^{MM}, Tm\text{-}1^{OE}/Tm\text{-}1^{OE}$ | T$_2$ transgenic | 3 | $0.5^B$ | $1.5^B$ | $0.5^B$ | $835.6^A \pm 207.2$ | 9 |
| | | $11^{MM}/11^{MM}, tm\text{-}1/tm\text{-}1$ | T$_2$ azygous control | 3 | $3.0^A$ | $2.9^A$ | $2.7^A$ | $617.0^A \pm 190.3$ | 1 |
| | TM-185 | $11^{MM}/11^{MM}, Tm\text{-}1^{OE}/Tm\text{-}1^{OE}$ | T$_2$ transgenic | 3 | $0.6^B$ | $0.9^B$ | $1.2^B$ | $260.9^A \pm 139.1$ | 14 |
| | | $11^{MM}/11^{MM}, tm\text{-}1/tm\text{-}1$ | T$_2$ azygous control | 3 | $3.0^A$ | $3.0^A$ | $2.8^A$ | $261.3^A \pm 46.7$ | 1 |
| | TM-186 | $11^{MM}/11^{MM}, Tm\text{-}1^{OE}/Tm\text{-}1^{OE}$ | T$_2$ transgenic | 3 | $0.9^B$ | $2.0^B$ | $2.1^B$ | $857.4^A \pm 77.4$ | 3 |
| | | $11^{MM}/11^{MM}, tm\text{-}1/tm\text{-}1$ | T$_2$ azygous control | 3 | $3.0^A$ | $3.0^A$ | $2.7^A$ | $428.6^A \pm 82.2$ | 1 |

The table presents average disease severity index (DSI) at 20, 30, and 46 DPI, average ToBRFV level at 30 DPI, *Tm-1* transcription range at 30 days post inoculation (DPI), and *Tm-1* fold-transcription in control plants (A) and in susceptible T$_2$ 'Moneymaker' plants overexpressing *Tm-1* compared to their azygous control counterparts (B). Results are presented as mean ± standard error; 11 represents the genetic source of the locus controlling tolerance on chromosome 11, where $^{MM}$ and $^{LA2825}$ signify the susceptible allele originating from 'Moneymaker' and LA2825, respectively; $Tm\text{-}1^{OE}$ symbolizes the introduced *Tm-1* overexpression transgene; BR denotes the number of biological repeats, each composed of five different T$_2$ or parental plants; average virus levels are presented as optical density (OD) ×1000. *Tm-1* transcription is presented as range and as fold change compared to the 'Moneymaker' genotype (A) or as compared to the azygous control genotype of each pair (B).

Means with different superscript letters are significantly different at $P < 0.05$, based on pair-wise comparisons using Student's *t*-test.

reductions ranging from 16% to 95% relative to wild-type VC554 levels (S8 Table). Following the self-fertilization of these T$_0$ plants, we obtained three T$_1$ progeny populations demonstrating segregation of the $Tm\text{-}1^{AS}$ construct. Three T$_1$ lines, designated TM-187, TM-188, and TM-189, were selected for in-depth analysis. These lines were subsequently used to develop transgenic and non-transgenic homozygous T$_2$ lines.

To assess the effect of *Tm-1* silencing on VC554 resistance, we conducted comparative analyses using three pairs of homozygous T$_2$ populations. Each pair comprised transgenic (carrying $Tm\text{-}1^{AS}$) plants and their non-transgenic (azygous) counterparts derived from a different T$_0$ progenitor. The results of these analyses are presented in Tables 4 and S9.

The T$_2$ population originating from the T$_1$ line TM-187 showed a strong and significant reduction of 83% in *Tm-1* expression level in the transgenic population compared to its non-transgenic azygous control. This reduction was associated with significantly higher virus-accumulation levels and disease symptoms in the transgenic plants compared to their azygous counterparts. In contrast, the transgenic T$_2$ plants originating from the T$_1$ line TM-188 did not display a reduction *Tm-1* expression compared to their azygous non-transgenic controls, indicating ineffective *Tm-1* silencing. As expected, this resulted in an insignificant effect on disease symptoms and virus-accumulation levels for the transgenic plants vs. their azygous control counterparts.

In the T$_2$ populations originating from the T$_1$ line TM-189, a reduction of 50% was observed in *Tm-1* expression in the transgenic plants compared to their non-transgenic counterparts. However, this non-significant reduction did not affect virus accumulation or symptom severity in the transgenic plants compared to their respective azygous controls.

**Table 4. Silencing *Tm-1* expression in the resistant VC554 genotype.**

| Original T$_1$ parental population | Description | Type | BR | DSI | ToBRFV level (OD, 405 nm) | Average *Tm-1* expression | Fold *Tm-1* transcription |
|---|---|---|---|---|---|---|---|
| TM-187 | $11^{VC554}/11^{VC554}$, *Tm-1*$^{AS}$/*Tm-1*$^{AS}$ | T$_2$ transgenic | 3 | 1.9$^A$ | 587$^A$±11 | 6.1$^A$±1.9 | 0.17 |
| | $11^{VC554}/11^{VC554}$, *Tm-1*/*Tm-1* | T$_2$ azygous control | 3 | 0.3$^B$ | 143$^B$±49 | 34.8$^B$±6.25 | 1 |
| TM-188 | $11^{VC554}/11^{VC554}$, *Tm-1*$^{AS}$/*Tm-1*$^{AS}$ | T$_2$ transgenic | 3 | 0.8$^A$ | 591$^A$±153 | 40.8$^A$±4.1 | 1.14 |
| | $11^{VC554}/11^{VC554}$, *Tm-1*/*Tm-1* | T$_2$ azygous control | 3 | 0.7$^A$ | 292$^A$±94 | 35.7$^A$±2.8 | 1 |
| TM-189 | $11^{VC554}/11^{VC554}$, *Tm-1*$^{AS}$/*Tm-1*$^{AS}$ | T$_2$ transgenic | 3 | 0.3$^A$ | 180$^A$±42 | 25.8$^A$±5.8 | 0.5 |
| | $11^{VC554}/11^{VC554}$, *Tm-1*/*Tm-1* | T$_2$ azygous control | 3 | 0.8$^A$ | 138$^A$±26 | 48.7 $^A$±13.9 | 1 |

The table presents average disease severity index (DSI), average ToBRFV level, *Tm-1* transcription range, and *Tm-1* fold-transcription in *Tm-1*-silenced VC554 T2 plants compared to their azygous control counterparts at 30 days post inoculation (DPI). Results are presented as mean ± standard error; $11^{VC554}$ represents the locus controlling tolerance on chromosome 11, originating from the resistant VC554 line; *Tm-1*$^{AS}$ symbolizes the introduced RNAi-silencing construct; BR denotes the number of biological repeats, each composed of five different T$_2$ or parental plants, Average virus levels are presented as optical density (OD) ×1000. *Tm-1* transcription is presented as range and as fold change compared to the azygous control genotype of each pair. Means with different superscript letters are significantly different at $P < 0.05$, based on pair-wise comparisons using Student's *t*-test.

## Whole-genome sequencing of the *Tm*-1 locus

To study the architecture of the *Tm-1* locus, validate non-synonymous nucleotide changes within and around the *Tm-1* CDS, and map novel genes linked to *Tm*-1, we used the fourth-generation Revio sequencing technology by PacBio (https://www.pacb.com/revio) to sequence the genomes of the three main genotypes examined in this study: the susceptible 'Moneymaker' line, the tolerant VC532 line, and the resistant VC554 line. Before the analysis, we generated highly accurate sequencing reads, followed by de novo assembly of chromosome 2, complemented by comprehensive gene-prediction analyses. Our DNA quality-control assessment showed that 90% of the genomic DNA fragments were longer than 10 kb, with 50% exceeding 30 kb, aligning with PacBio's recommended standards. Using the Revio SMRT cell, we generated 70.7 Gb of data. The HiFi read lengths achieved an N50 of 15.2 kb, with a mean Q-value of 33.

All sequences were initially subjected to de novo assembly to reconstruct chromosome 2 for the three genomes (VC554, VC532, and Moneymaker). Gene-prediction analysis, performed using these assemblies, did not reveal any novel gene in the proximal genomic region of *Tm-1* compared to the reference genome. However, examining the CDS of *Tm-1* across the three genomes tested showed that the resistant genotype VC554 carries two distinct copies of *Tm-1* separated by approximately 27 kb. The first copy, referred to as *Tm-1*$^{VC554-1st}$, perfectly matched the nucleotide sequence of the well-documented resistant *Tm-1* gene derived from the ToMV-resistant genotype GCR237 (*Tm-1*$^{GCR237}$) [3]. In contrast, the second copy, referred to as *Tm-1*$^{VC554-2nd}$, shared 97% nucleotide sequence identity with *Tm-1*$^{GCR237}$, resulting in 95% amino acid sequence identity.

Discovery of the two *Tm-1* gene copies prompted us to analyze their expression differences (if any) in the VC554 genotype. We conducted a quantitative PCR (qPCR) analysis using two sets of primers: *Tm-1_1stq* specific to *Tm-1*$^{VC554-1st}$ and *Tm-1_2ndq* specific to *Tm-1*$^{VC554-2nd}$ (Table 5). Expression levels of the *Tm-1*$^{VC554-2nd}$ gene copy were significantly lower than those of the *Tm-1*$^{VC554-1st}$ copy. In fact, the expression level of the second copy was undetectable in some samples (S10 Table).

In contrast to the resistant genotype, sequence analysis of this region in the tolerant VC532 genotype revealed only a single copy of the gene, with a single non-synonymous nucleotide change compared to the reference *tm-1* allele sequence identified in the susceptible GCR26 genotype (*tm-1*$^{GCR26}$) by Ishibashi et al. (2007) [3]. This nucleotide change resulted in a His to Arg change at position 337. Surprisingly, this single-nucleotide polymorphism aligned with the resistance allele *Tm-1*$^{GCR237}$. As expected, the susceptible 'Moneymaker' genotype's CDS was identical to the susceptible allele sequence of *tm-1*$^{GCR26}$. Multiple sequence alignment of the *Tm-1* CDSs and their corresponding amino acids across

the three genomes is shown in the online supplementary material (S1 and S2 Figs). The discovery of two *Tm-1* copies in VC554, coupled with the fact that *Tm-1* was initially identified in *Solanum habrochaites* [10], led us to examine the chromosome 2 sequences of the *S. habrochaites* reference genome. We found that the *S. habrochaites* ZY59 reference genome ([https://www.ncbi.nlm.nih.gov/datasets/genome/GCA_027704765.1](https://www.ncbi.nlm.nih.gov/datasets/genome/GCA_027704765.1), accessed March 6, 2025) also carries two *Tm-1* copies—7.5 kb and 6.3 kb long—separated by 11.8 kb.

## Discussion

ToBRFV is a rapidly spreading tobamovirus that severely impacts tomato production and fruit quality [19,33]. The most effective, economically viable, and sustainable approach to suppressing ToBRFV's aggressive replication and inoculation is to identify and introgress resistant genes into commercial breeding lines [34,35].

Our previous work showed that a locus on chromosome 11 controls tolerance to ToBRFV in the VC532 genotype, while the resistance trait in the VC554 genotype results from introducing the locus controlling tolerance on chromosome 11 and the *Tm-1* locus on chromosome 2 [32]. This study aimed to further elucidate the role of *Tm-1* in ToBRFV resistance and its interaction with the tolerance locus on chromosome 11.

Our findings revealed that the *Tm-1* locus, in combination with the tolerance locus on chromosome 11, is essential for conferring ToBRFV resistance. Overexpressing the *Tm-1* gene in the tolerant VC532 genotype resulted in a significant reduction in viral accumulation compared to non-transgenic VC532 plants, hence conferring a very high level of resistance to ToBRFV. Conversely, *Tm-1* overexpression in the susceptible 'Moneymaker' genotype only delayed symptom appearance without significantly affecting viral accumulation, indicating that *Tm-1* interacts genetically with the locus controlling tolerance on chromosome 11 to achieve effective resistance against ToBRFV. The elevated symptom severity observed

**Table 5. Primers used in this study.**

| Primer name | Forward (F) and reverse (R) primer sequences (5'→3') |
|---|---|
| *Tm-1OESalI_F* | GTCGACATGGCAACTGCACAGAG |
| *Tm-1OENotI_R* | GCGGCCGCGTCACTCCATAGAAATAG |
| *Tm-1OE_F* | GAGATCCAGTCTTAACAGCTTCTCC |
| *Tm-1OE_R* | ACTGAAGGAAACAATACCAAGTCTG |
| *35S_F* | CAAGACCCTTCCTCTATATAAG |
| *Intron_R* | CTAGTATATCATCTTACATGTTCG |
| *Tm-1ASXhoI_F* | ACACTCGAGTTCCTCTCCGAGCATGTG |
| *Tm-1ASEcoRI_R* | GCTGAATTCTCCACTACCCCCAAGGCC |
| *Tm-1ASXbaI_F* | ACATCTAGATTCCTCTCCGAGCATGTG |
| *Tm-1ASBamHI_R* | GCTGGATCCTCCACTACCCCCAAGGCC |
| *Tm-1_F* | TCTCACCATTCTCACACTGAGTTAC |
| *Tm-1_R* | ACTGCCGGAAACAATACCAAGTCG |
| *Tm-1q_F* | ACCCCATATGCTTTCTGCCC |
| *Tm-1q_R* | GGGGAAGATATAGGGCCTCC |
| *Tm-1_1$^{st}$q_F* | AGCTTCGTTTCCTCTCCGAGC |
| *Tm-1_1$^{st}$q_R* | CAAACGTGCCCATAGTTTCTT |
| *Tm-1_2$^{nd}$q_F* | AGCTTCGTTTCCTCTCCCAAT |
| *Tm-1_2$^{nd}$q_R* | GAAGCTGGACCACAGATTCTC |
| *tm-1$^{VC532}$_F* | GTGCATCCAATGCAGACACG |
| *tm-1$^{VC532}$_R* | ACGCTTAATACTCACCTGTTACTG |
| *18s_F* | GCGACGCATCATTCAAATTTC |
| *18s_R* | TCCGGAATCGAACCCTAATTC |

[https://doi.org/10.1371/journal.pgen.1011725.t005](https://doi.org/10.1371/journal.pgen.1011725.t005)

in the line LA2825 harboring *Tm-1* compared to the $T_2$ transgenic populations, can be attributed to the higher expression levels of *Tm-1* in the transgenic plants relative to its normal expression levels in LA2825.

Our efforts to interfere with *Tm-1* expression levels in the resistant VC554 genotype by RNAi revealed varying effects, depending on the extent of expression reduction. A two-fold decrease in *Tm-1* expression did not alter resistance. However, a five-fold decrease significantly affected the resistant plants, characterized by elevated virus levels and a substantial increase in symptom severity compared to the non-transgenic azygous VC554 control.

Design and analysis of our *Tm-1$^{AS}$* silencing construct were conducted prior to obtaining the PacBio sequencing results. Consequently, during the initial stages of our study, we were unaware of the existence of a second copy of *Tm-1* in the VC554 genotype (*Tm-1$^{VC554-2nd}$*). We examined whether sequence differences between the two *Tm-1* copies in the VC554 genotype could affect the efficiency of *Tm-1* silencing. The RNAi construct was designed to target a 303-bp sequence, which exhibited a 10% variation between the two copies. While previous studies have demonstrated that a single RNAi construct can silence multiple homologous genes simultaneously [36,37], we could not confirm whether our construct effectively silenced both *Tm-1* copies. Importantly, the *Tm-1q* primers (Table 5) used for quantifying *Tm-1* expression were designed to complement sequences in both *Tm-1* copies, ensuring the detection of total *Tm-1* transcript abundance. Subsequent qPCR analysis using primers specific to the second *Tm-1* copy (*Tm-1_2ndq*, Table 5) revealed that this copy (*Tm-1$^{VC554-2nd}$*) exhibits negligible expression levels in non-transgenic lines (S10 Table), suggesting its limited functional relevance. Given the extremely low expression of the second copy in non-transgenic lines, our silencing strategy primarily affected the first copy, which appears to be the main functional variant of *Tm-1* in this genotype.

Given that the VC554 genotype carries the locus controlling tolerance on chromosome 11, we hypothesized that silencing *Tm-1* expression would convert the resistant VC554 genotype to a tolerant one, characterized by low symptom severity and high virus accumulation. However, surprisingly, *Tm-1* silencing in VC554 rendered the resistant genotype VC554 susceptible, although symptoms levels in the transgenic plants were still less severe than those observed in susceptible plants across all experiments throughout the study. Two potential mechanisms might explain the observed susceptibility phenotype with reduced symptom severity. First, even at high levels of *Tm-1* silencing, a certain degree of interaction is maintained, sufficient to reduce symptom severity but not the viral load. Second, these findings suggest that normal expression levels of the recessive *tm-1* gene might be essential for achieving complete ToBRFV tolerance. Thus, we propose that ToBRFV resistance requires the interaction of the dominant allele of *Tm-1* with the locus on chromosome 11, whereas interaction of the recessive allele *tm-1* with the same locus on chromosome 11 induces tolerance to the virus. The limited tolerance observed in transgenic plants originating from TM-187 may reflect the independent action of the chromosome 11 locus, unaffected by the presence of either *Tm-1* allele.

Our hypothesis that normal expression levels of the recessive *tm-1* gene might be essential for obtaining ToBRFV tolerance is supported by a previous study [38] in which tomato plant resistance to non-host tobamoviruses was suggested to be based on an inhibitory interaction between viral replication proteins and the host cellular protein encoded by *tm-1*.

Interestingly, our PacBio sequencing analysis revealed that the *tm-1* protein in the tolerant VC532 genotype, referred to as *tm-1$^{VC532}$*, differs by a single amino acid from the susceptible *tm-1$^{GCR26}$* sequence previously described by Ishibashi et al. [3], and carried by the 'Moneymaker' genotype. Therefore, the possible involvement of this novel *tm-1* gene variant in maintaining tolerance in VC532 cannot be dismissed. However, a two-way analysis of 220 $F_2$ plants resulting from a cross between VC532 and 'Moneymaker' [32] was performed utilizing a specific co-dominant marker. This marker, based on a SNP that differentiates between the *tm-1* alleles and introduced a restriction site for the *Rsa*I enzyme. The genomic region adjacent to *tm-1* and flanked by *tm-1$^{VC532}$* primers was amplified by PCR (Table 5), and subsequent restriction analysis using *Rsa*I facilitated alleles differentiation. This analysis showed that *tm-1$^{VC532}$* has no additive effect on symptom severity [*P(F) = 0.74*] and shows no significant interaction with the locus on chromosome 11 [*P(F) = 0.98*]. As expected, the QTL on chromosome 11 demonstrated a highly significant effect [(*P(F) = 2.84 × 10$^{-75}$*], confirming our previous findings [32],(S11 Table). This lack of effect of *tm-1$^{VC532}$* on symptom severity is in agreement with our previous results showing that

this allele is also ineffective at controlling ToBRFV resistance. Based on these findings, we can conclude that the novel *tm-1* allele of the tolerant genotype VC532 (*tm-1^VC532*) does not contribute to the tolerance mechanism in VC532, and is ineffective at controlling resistance to the virus.

A recent study by Kubota et al. [39] demonstrated that ToBRFV can evolve within *Tm-1*-harboring plants, potentially circumventing the *Tm-1* effect on resistance. This finding highlights the virus adaptability and demonstrates the need for continued studies into the broader impact of viral evolution on resistance mechanisms.

In conclusion, our study offers significant insights into the genetic mechanisms underlying resistance and tolerance to ToBRFV in tomatoes. It provides compelling evidence for *Tm-1's* involvement in ToBRFV resistance, specifically through its interaction with a tolerance locus on chromosome 11. This interaction is essential for achieving durable resistance, characterized by both reduced symptom severity and low virus accumulation. The unexpected susceptibility observed in *Tm-1*-silenced VC554 plants suggests a more complex mechanism underlying control of the tolerance trait than previously assumed, leading us to propose that normal expression levels of the recessive *tm-1* allele may be essential for achieving complete tolerance to ToBRFV, potentially through interaction with the chromosome 11 locus.

Further research is essential to fully elucidate the specific molecular mechanisms underlying the interactions between each of the two *Tm-1* gene alleles and the locus on chromosome 11. Understanding these genetic interactions in relation to viral dynamics will be crucial for developing robust and durable ToBRFV-resistant tomato varieties, thereby enhancing crop-protection strategies against this economically significant pathogen.

## Materials and methods

### Plant material and resource populations

Seeds of tomato accessions were acquired from three sources: the Tomato Genetics Resource Center (TGRC) at the University of California, Davis, USA (https://tgrc.ucdavis.edu/), the laboratory of Professor Dani Zamir at the Hebrew University in Rehovot, Israel, and the Volcani Center seed collection. These genotypes included *S. lycopersicum* cv. Moneymaker (LA2706) as a susceptible control genotype, a *Tm-1/Tm-1* genotype in cv. Moneymaker (LA2825), the tolerant genotype *S. lycopersicum* var. *cerasiforme* (VC532), and the resistant genotype, cultivated tomato accession VC554.

### Development and genotyping of an F$_3$ population segregating for *Tm-1*

An F$_3$ population segregating for the *Tm-1* locus while fixed for the locus on chromosome 11 originating from the tolerant VC532 genotype was developed by self-pollination of an F$_2$ plant. This F$_2$ plant, homozygous for the VC552-derived chromosome 11 locus and heterozygous for the *Tm-1* locus, was obtained from our previously reported allelic test [32]. Genotyping of the F$_3$ plants at the *Tm-1* locus was conducted using PCR amplification with primers specific to the *Tm-1* gene: *Tm-1_F* and *Tm-1_R* (Table 5). The resulting PCR products were digested with *Stu*I endonuclease to differentiate between genotypes carrying the resistant allele and those carrying the susceptible allele. The digested fragments were visualized by electrophoresis on a 2% agarose gel.

### Virus maintenance and plant inoculation

The ToBRFV isolate (GenBank Acc. No. KXG619418) was maintained on *S. lycopersicum* cv. Moneymaker plants carrying the ToMV resistance gene *Tm-2²* (LA3310). Plants were maintained in an insect-proof greenhouse, with virus cultures perpetuated through serial mechanical inoculations at 21- to 28-day intervals. For each inoculation, infected leaf tissue was homogenized in 0.01 M phosphate buffer (pH 7.0). The viral homogenate was mechanically applied to carborundum-dusted leaves of test plants. Following inoculation, leaves were rinsed to remove residual carborundum, and plants were transferred to a temperature-controlled greenhouse maintained at 18–25°C under natural photoperiod conditions.

Seeds were sown in standardized germination trays (8 × 16 matrix configuration) containing 40 mL of commercial grow-ing medium per cell (Hishtil, Nehalim, Israel). Individual cells served as discrete experimental units for seedling develop-ment and subsequent viral challenge.

### Disease severity scoring

ToBRFV-induced symptoms were evaluated 30 DPI and at later times, according to a DSI consisting of: (0) no visible symptoms, with inoculated plants displaying average growth and development, indistinguishable from non-inoculated plants; (1) mild mosaic pattern observed on the apical leaf; (2) distinct mosaic pattern on the apical leaves leaf; (3) pronounced mosaic pattern accompanied by notable elongation or deformation of the apical leaves [32]. Fig 1 displays representative images of each DSI level.

### Enzyme-linked immunosorbent assay (ELISA) to evaluate viral levels

Indirect ELISA analyses were conducted on plant leaves using specific antibodies against ToBRFV (generously provided by Dr. A. Dombrovsky, ARO, Rishon LeZion, Israel), based on the protocols established by Luria et al. [16] and Koenig [40]. Two 1-cm diameter discs were taken from each tested plant's fourth and fifth leaves. Samples were collected 30 DPI, ground in a coating buffer (Agdia), and incubated for 3 h at 37°C with a 1:5000 dilution of the anti-ToBRFV antiserum. For the subsequent detection step, samples were treated with alkaline phosphatase-conjugated goat anti-rabbit IgG (Sigma, Steinheim, Ger-many) for 3 additional hours at 37°C. Para-nitrophenylphosphate (Sigma) substrate was used at 0.6 mg mL$^{-1}$. The developing color was recorded by a Multiskan FC microplate photometer (Thermo Fisher Scientific, Waltham, MA, USA) at 405 nm.

### Genomic DNA extraction and PCR

Genomic DNA was extracted from individual plants according to Fulton et al. (1995) [41]. PCR primers were designed using the NCBI primer-blast tool (https://www.ncbi.nlm.nih.gov/tools/primer-blast) and are presented in Table 5. The PCRs were carried out in a 20 µL volume, comprised of 10 µL Hy-taq ready mix by HyLabs (https://www.hylabs.co.il), 1 µL of each relevant primer (10 µM), and 8 µL of ultra-pure H$_2$O. PCR conditions were: initial denaturation at 94°C for 3 min, fol-lowed by 35 cycles of 94°C for 30 s, 58–60°C for 30 s (depending on primer characteristics), and 72°C for 1 min. The final elongation was at 72°C for 10 min.

### High-molecular-weight (HMW) DNA extraction, quality assessment, library preparation, and PacBio sequencing

HMW plant DNA was extracted from 1.5 g of young leaves using the NucleoBond HMW DNA kit (Macherey-Nagel, Duren, Germany). Quantity and purity of the HMW DNA were assessed using Qubit Fluorometric Quantification (Thermo Fisher Scientific, Wilmington, NC, USA) and a NanoDrop 1000 spectrophotometer (Thermo Fisher Scientific), respectively. DNA-fragment size distribution was analyzed using a FEMTO pulse device (Agilent, Santa Clara, CA, USA).

HMW DNA was sent to the DNA Technologies Core facility at UC Davis Genome Center (https://dnatech.genomecen-ter.ucdavis.edu/expression-analysis-core/). The DNA quality was verified, and sequencing libraries were constructed and subjected to PacBio High Fidelity (HiFi) sequencing using the fourth-generation sequencing technology Revio (https://www.pacb.com/revio).

### Bioinformatics analysis of revio sequence data

Raw sequence data generated from the Revio sequencer were assembled by hifiasm (v0.16.1) with default parameters [42]. Genome-assembly completeness was assessed using Benchmarking Universal Single-Copy Orthologs against single-copy orthologs in the Viridiplantae lineage (BUSCO v5.0.0) [43], and contiguity was assessed using QUAST (ver-sion 5.0.2) [44].

Repeat elements were detected de novo by RepeatModeler version v1.0.11 and masked by RepeatMasker version 4.0.9_p2 [45]. Evidence-based gene prediction was performed using OmicsBox 2.0.36. The predicted proteins were used as a query term to search the NCBI non-redundant (nr) protein database using the DIAMOND program [46], and functional annotation was performed in OmicsBox 2.0.36.

Genome sequences were compared using Mummer software [47] set at NUCmer on the reference *S. lycopersicum* cv. Heinz 1706 SL4.0 genome. The cords output of NUCmer was analyzed to extract the homology scaffolds to chromosome 2.

### Relative transcription level of *Tm-1*

*R*elative levels of *Tm-1* transcription were determined by real-time qPCR. Total RNA was extracted from 100 mg of young leaf tissue with the NucleoSpin RNA plant isolation kit (Macherey-Nagel). RNA purity was assessed using a spectrophotometer (Thermo Fisher Scientific). Total RNA (1 µg) was used as the template for first-strand cDNA synthesis with the qScript cDNA Synthesis Kit by Quantabio (Beverly, MA, USA). *Tm-1q* primers (Table 5) for qPCR were designed using the NCBI primer-blast online tool (https://www.ncbi.nlm.nih.gov/tools/primer-blast) to form a 221-bp amplicon.

Each qPCR was carried out in a total volume of 12 µL consisting of 6 µL ABsolute Blue qPCR Master Mix with ROX (Thermo Fisher Scientific), 1.2 µL of each primer (0.15 µM), 0.6 µL of ultra-pure $H_2O$, and 3 µL of cDNA template. The reactions consisted of: 40 cycles at 95°C for 15s, 60°C for 30s, and 72°C for 30s. All reactions were conducted in a StepOne Plus Real-Time PCR System (Applied Biosystems, Waltham, MA, USA) using a 96-well plate. Data were analyzed using StepOne software v 2.3 (Applied Biosystems). Relative expression of the *Tm-1* transcript was determined with *18S* ribosomal RNA (rRNA) as the reference gene. Amplification was conducted using the *18s_F* and *18s_R* primers (Table 5). The calculation was performed using the formula: $2^{-(Ct\_Tm-1 - Ct\_18S)}$, where Ct represents the cycle number at which the fluorescence crosses a predefined threshold. The data are presented as fold change in gene expression standardized to the *18S* rRNA reference gene and compared to either the non-transgenic or non-transgenic azygous controls.

### Overexpression of *Tm-1*

Total cDNA from LA2825 was used in a PCR with *Tm-1OESalI_F* and T*m-1OENotI_R* primers (Table 5) to develop a *Tm-1* CDS clone. This clone, featuring *Sal*I and *Not*I restriction sequence targets, was inserted into a pBIN vector containing neomycin phosphotransferase II, CaMV *35S* promoter, *Sal*I–*Not*I sites, and NOS terminator, forming the pBIN*Tm-1*-LA2825 expression cassette. Cotyledon cuttings were transformed using *Agrobacterium tumefaciens* strain GV3103, according to Azari et al. [48].

$T_0$ plants overexpressing *Tm-1* were generated in VC532 and 'Moneymaker' backgrounds and self-pollinated to produce $T_1$ populations segregating for the *Tm-1*$^{OE}$ transgene. Individual $T_1$ plants were distinguished as either transgenic or non-transgenic using PCR analysis with specific *Tm-1OE* primers (Table 5). These plants were further self-pollinated to create $T_2$ populations.

### Reducing *Tm-1* expression level

The pHannibal vector [49], designed to express both sense and antisense fragments of the gene, was constructed through a two-step process. Initially, a 303-bp fragment of the *Tm-1* cDNA gene (spanning nucleotides 76–378 in the cDNA sequence) was PCR-amplified, utilizing primers *Tm-1ASXhoI_F* and *Tm-1ASEcoRI_R*, and introducing *Xho*I and *EcoR*I restriction sites, respectively (Table 5). The fragment was inserted into the *Xho*I and *EcoR*I sites in the sense-oriented pHannibal region. The identical 303-bp gene fragment was then PCR-amplified using the *Tm-1ASXbaI_F* and *Tm-1ASBamHI_R* primers, encompassing an *Xba*I and *BamH*I site, respectively (Table 5). This fragment was integrated into the specific *Xba*I and *BamH*I sites found in the backward-oriented region of pHannibal, resulting in the creation of pHannibal-*Tm-1*. To create a binary vector, pHannibal-*Tm-1* was inserted under the CaMV *35S* promoter and the OCS

terminator into the *Not*I site of the pBIN vector. Transformations were conducted on cotyledon cuttings from VC554 using *A. tumefaciens* strain GV3101, as previously described [48].

All $T_0$ plants were tested by PCR to determine whether they were transgenic using *35S_F* and *Intron_R* specific primers (Table 5). Three $T_0$ plants were self-pollinated to obtain three $T_1$ populations segregating for the *Tm-1* RNAi construct. We created $T_2$ populations through self-pollination. For each of the three $T_0$ plants, we selected 10 transgenic $T_1$ plants and 2 non-transgenic $T_1$ plants. These selected plants were then self-pollinated to produce the $T_2$ generation.

## Characterization of transgenic plants

Integration of the three transformation constructs created in this study was confirmed by PCR using DNA extracted from individual transformed plants as templates. For *Tm-1*-overexpressing plants, we utilized the transgenic *Tm-1*-overexpression primers *Tm-1OE* (Table 5), each located on a different exon with one intron between them. Consequently, these primers are expected to amplify a genomic fragment of 971 bp. In *Tm-1$^{OE}$*-transgenic plants, these primers will also amplify the incorporated *Tm-1* CDS (exons only, without the intron), resulting in an additional 400-bp fragment. To verify the integration of the *Tm-1* RNAi (*Tm-1$^{AS}$*) construct into VC554 plants, we conducted PCR using the forward primer *35S_F* (Table 5), which targets the CaMV promoter, and the reverse primer *Intron_R* (Table 5), which binds between the sense and antisense regions of the pHannibal vector. This amplification yielded a 623-bp PCR product.

## Development of homozygous transgenic and non-transgenic $T_2$ populations

We used a progeny-testing approach to obtain homozygous $T_2$ populations for plants overexpressing *Tm-1* in VC532 and in 'Moneymaker', as well as *Tm-1* silencing in VC554. Transgenic $T_0$ plants were self-pollinated to produce $T_1$ populations segregating for the transgene. Individual $T_1$ plants were identified as either transgenic or non-transgenic using the appropriate primers by PCR and allowed to self-pollinate to obtain $T_2$ populations. Thirty-two plants of each transgenic and non-transgenic $T_2$ population were genotyped, identifying and validating homozygous transgenic and non-transgenic $T_2$ populations for each of the three transgenes. Homozygous $T_2$ plants were then used for the experiments, where each transgenic and non-transgenic homozygous $T_2$ population was represented by three biological replicates, each consisting of five plants.

## Sequence-alignment analysis

Nucleotide- and amino acid-sequence alignments of *Tm-1* (PacBio) sequenced variants were analyzed and visualized using the Multiple Sequence Alignment (msa) package [50] and ggmsa package [51] in R software.

## Statistical analyses

Analyses of variance assessed differences in average virus levels and DSI. All analyses were conducted with the JMP Pro 15 statistical discovery software (SAS Institute Inc., Cary, NC, USA). Differences among means are presented as different superscript letters that represent statistically significant differences between mean values ($P < 0.05$), based on the Tukey–Kramer Honestly Significant Difference (HSD) test [52] or Student's *t*-test.

## Supporting information

**S1 Fig.  *Tm-1* coding sequence nucleotide alignment.**
(DOCX)

**S2 Fig.  *Tm-1* amino acid sequence alignment.**
(DOCX)

**S1 Table. Disease Severity Index (DSI) metrics show the number of plants in each DSI in the experiment analyzing the Association of *Tm-1* with Resistance phenotype in plants carrying the tolerance locus.**
(DOCX)

**S2 Table. *Tm-1* transcription fold across VC532-*Tm-1*-OE $T_0$ transgenic plants.**
(DOCX)

**S3 Table. Disease Severity Index (DSI) metrics show the number of plants in each DSI in the experiment analyzing overexpression of *Tm-1* in the tolerant genotype VC532.**
(DOCX)

**S4 Table. *Tm-1* transcription fold across Moneymaker -*Tm-1*-OE $T_0$ transgenic plants.**
(DOCX)

**S5 Table. Disease Severity Index (DSI) metrics show the number of plants in each DSI at 20 DPI in the experiment analyzing overexpression of *Tm-1* in the susceptible genotype Moneymaker.**
(DOCX)

**S6 Table. Disease Severity Index (DSI) metrics show the number of plants in each DSI at 30 DPI in the experiment analyzing overexpression of *Tm-1* in the susceptible genotype Moneymaker.**
(DOCX)

**S7 Table. Disease Severity Index (DSI) metrics show the number of plants in each DSI at 46 DPI in the experiment analyzing overexpression of *Tm-1* in the susceptible genotype Moneymaker.**
(DOCX)

**S8 Table. *Tm-1* transcription fold across VC554-*Tm-1*-AS $T_0$ transgenic plants.**
(DOCX)

**S9 Table. Disease Severity Index (DSI) metrics show the number of plants in each DSI in the experiment analyzing the reduction of *Tm-1* expression in the resistant genotype VC554.**
(DOCX)

**S10 Table. *Tm-1* 1st and *Tm-1* 2nd expression level in *Tm-1* harboring genotypes.**
(DOCX)

**S11 Table. Disease Severity Index (DSI) metrics show the number of plants in each DSI in the $F_2$ population used for the two-way analysis to evaluate *tm-1*VC532 effect on symptom severity.**
(DOCX)

## Acknowledgments

The authors wish to thank Aviv Dombrovsky (Volcani Center) for the use of anti-ToBRFV antibodies developed in his laboratory. The presented data are part of the Ph.D. thesis of AZ, supervised jointly by IL and ML.

## Author contributions

**Conceptualization:** Ilan Levin, Moshe Lapidot.

**Data curation:** Avner Zinger, Adi Doron-Faigenboim, Dana Gelbart.

**Formal analysis:** Avner Zinger, Adi Doron-Faigenboim, Ilan Levin.

**Funding acquisition:** Ilan Levin, Moshe Lapidot.

**Investigation:** Avner Zinger, Dana Gelbart.

**Methodology:** Ilan Levin, Moshe Lapidot.

**Project administration:** Ilan Levin.

**Resources:** Ilan Levin, Moshe Lapidot.

**Software:** Adi Doron-Faigenboim.

**Supervision:** Ilan Levin, Moshe Lapidot.

**Validation:** Avner Zinger, Dana Gelbart.

**Visualization:** Avner Zinger.

**Writing – original draft:** Avner Zinger, Adi Doron-Faigenboim, Ilan Levin, Moshe Lapidot.

**Writing – review & editing:** Avner Zinger, Ilan Levin, Moshe Lapidot.

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
