## [Decision Letter · Decision Letter 0]

3 Mar 2025

PGENETICS-D-25-00121

Contribution ofthe Tobamovirus resistance gene Tm-1 to control of ToBRFV resistance in tomato

PLOS Genetics

Dear Dr. Lapidot,

Thank you for submitting your manuscript to PLOS Genetics. After careful consideration, we feel that it has merit but does not fully meet PLOS Genetics's publication criteria as it currently stands. Therefore, we invite you to submit a revised version of the manuscript that addresses the points raised during the review process.

As you will see from the comments of the reviewers, they appreciate the proof for the cooperation between the Tm-1 gene and a tolerance-conferring locus on chromosome 11 for resistance against ToBRFV. While they like the design of the study and the main findings, they found several issues that need to be addressed in a substantially revised version. Importantly, all data need to be made accessible to the reader, as the journal policy does not allow a statement like “data not shown” (appears 5 x). Make sure that there is sufficient information to allow a broad readership to follow the story. Please also follow the suggestion to provide informative images to categorize the symptom severity. In addition, there are several specific issues listed that need correction or consideration, by changes in the manuscript and by answering to the reviews.

Please submit your revised manuscript within 60 days May 02 2025 11:59PM. If you will need more time than this to complete your revisions, please reply to this message or contact the journal office at plosgenetics@plos.org. Please include the following items when submitting your revised manuscript:

We look forward to receiving your revised manuscript.

Kind regards,

Ortrun Mittelsten Scheid

Academic Editor

PLOS Genetics

Aimée Dudley

Editor-in-Chief

PLOS Genetics

Aimée Dudley

Editor-in-Chief

PLOS Genetics

Anne Goriely

Editor-in-Chief

PLOS Genetics

**Additional Editor Comments (if provided):**

**Journal Requirements:**

**Reviewers' comments:**

Reviewer's Responses to Questions

**Comments to the Authors:**

Reviewer #1: The authors have found that a locus controlling tolerable to the tobamovirus tomato brown rugose fruit virus can work synergistically with a well known but relatively weak tobamovirus resistance gene Tm-1 to provide resistance to the virus. Although incremental in nature, the work may pint a way towards aiding in the control of the growing pandemic of tomato brown rugose fruit virus.

Other comments

Title - name the virus in full. No reason why readers of a genetics journal will know what this is. Throughout the manuscript there is an inherent assumption that readers are already tobamovirus experts and a lack of clarity in exposition.

Line 110. 'Levitzky et al. [21] also discovered... can spread ToBRFV' implies the bees transmitted the virus to other plants and acted as vectors. However, I dio not think that any evidnce for this was presnted - only that bees became contaminated with viable inouclum, which is not the same thing as acting as a vector.

Please correct.

line 117. Please add dates of access for all URLs

Data presentation is unsatisfactory, often unclear, and over-assumes background knowledge of the readership:

Table 1 Line 176 and elsewhere throughout the manuscript. With respect to a disease severity index, since this is a subjective measure, it would be helpful for the readership to be able to see examples 0 – 4 levels of symptoms (is it just appearance or stunting? etc.).

Statistically, I do not think it is valid to have standard errors around subjective values, although I know it is often done, but is really not necessary. It would be more useful to know what proportion of plants in each group showed each DSI value. The description of the DSI values in materials and methods will not help anyone except an expert in tobamoviral infections of tomato (at line 180.... the line numbering in the ms is all over the place). Again, try and consider who your audience is here.

The raw data is not available as supp data: I would assume this is a requirement for publication. This applies to other tables in the manuscript.

Similarly, just mentioning 'optical density' with saying that is ELISA (?) OD at what wavelength etc. does not help the clarity for readers who are not plant virologists (even then, some clarity is helpful since none of us are mind-readers). The authors are trying to publish in Pols Genee tics, not Annals of Applied Biology or Plant Disease and should tke this into account.

Table 2 . Same comments about DSI as made re Table 1.

What readers probably want to see is a good collection of representative images- there's no limit as to what you put in supplementary files. Why not use this opportunity to convince, persuade and impress the readers of the work's importance?

Page 17. Table 3 not Tables 3. Again, the DSI with statistics is not helpful (and possibly not statistically valid - see earlier comments) , previously mentioned issues with ELISA data, and it is not clear how many plants were involved and therefore what the real statistical power was here. Table 4 etc similar points.

Reviewer #2: The manuscript presents an important and well-executed study on the genetic basis of resistance to tomato brown rugose fruit virus (ToBRFV), a pathogen of significant concern for tomato production worldwide. The research builds upon previous findings, providing compelling evidence of the interaction between the Tm-1 gene and a tolerance-conferring locus on chromosome 11. The experimental approach is well-designed, employing overexpression and RNAi-silencing strategies to dissect the role of Tm-1 in resistance. The results are clear and contribute valuable insights into the molecular mechanisms underlying resistance and tolerance to ToBRFV.

The discussion effectively integrates the findings within the broader context of viral resistance in tomato, offering a logical interpretation of the results and highlighting their potential applications in breeding programs. The manuscript is well-structured and clearly written, facilitating the comprehension of the study’s implications.

Only minor formal issues should be addressed. The authors should carefully review the use of upper/lower case and italics when referring to virus names, ensuring consistency with standard nomenclature. Additionally, a thorough proofreading will help correct any minor typographical errors that may remain. These minor revisions should be easily addressed and will further improve the manuscript’s clarity.

Overall, this is a strong and relevant study that will be of great interest to researchers working on plant-virus interactions and resistance breeding.

Reviewer #3: Authors (Zinger et al.) previously identified a locus on chromosome 11, in combination with a second locus mapped closely to the Tm-1 region on chromosome 2 from a tomato brown rugose fruit virus (ToBRFV)-resistant line (VC554). However, the Tm-1 locus was missing from the ToBRFV-tolerant line (VC532) but have the same locus on chromosome 11.

In the present study, authors conducted various experiments to determine the role of Tm-1 for its contribution to ToBRFV resistance. In evaluating 60 F3 lines that were segregating for ToBRFV resistance, they confirmed again the association of the Tm-1 locus in combination with the chromosome 11 locus for ToBRFV resistance.

Using transgenic tomato plants over-expressing the Tm-1 in the tolerance or the susceptible genotype background, they determined that the Tm-1 in combination with the chromosome 11 locus in the tolerance genotype achieved a high level of resistance to ToBRFV. However, the Tm-1 gene alone in the susceptible genotype yielded only some level of tolerance to ToBRFV without significant reduction in the virus titer.

Silencing of the Tm-1 gene in the resistance genotype resulted a diverse response to ToBRFV infection. Although one plant with Tm-1 silencing, two others did not, these phenomena may be due to an incomplete silencing of the Tm-1 gene. Genome resequencing revealed the presence of two copies of the Tm-1 gene in the resistance genotype, but only one copy of the tm-1 gene in the tolerance genotype.

Token together, these results offered strong evidence to support the contribution of the Tm-1, in combination with the locus in chromosome 11 resulting in the ToBRFV resistance.

However, following comments and suggestions would need to be considered in revisions.

1. All the data on (data not shown) need to be included in the complementary materials.

2. All the table headings should be with one sentence only. The second sentence should be moved down to the footnote section.

3. Line 51. Tomato (Solanum lycopersicum) is… to Tomato (Solanum lycopersicum L.) is…

4. Line 113. There are more countries than those listed here, it may be good to list EPPO for more comprehensive worldwide distribution (https://gd.eppo.int/taxon/tobrfv).

5. Line 123. Missing a citation of Jaiswal et al., 2024, https://www.mdpi.com/2223-7747/13/5/581

6. Page 15, on symptom severity, need to provide symptom photos in the supplementary materials. (the line number was not continuing from this page)

7. Page 15, on the LA2825 (Tm-1/Tm-1 genotype in cv. Moneymaker), why the transgenic Tm-1 plants would yield milder symptoms than the LA2825 plants with homozygous for the Tm-1 gene?

8. Page 26, line 77: change “we cloud not confirm” to “we could not confirm”.

9. Page 26, line 83: change given to Given.

10. Page 29, line 155: What is the VC552 genotype? Need to explain this line was derived from a cross between VC532 and VC554.

11. Page 41, Lines 380-381, the reference 12 is not complete.

**Have all data underlying the figures and results presented in the manuscript been provided?**

Reviewer #1: **No: ** Raw data used in tables should be included in supp data

Reviewer #2: Yes

Reviewer #3: **No: ** All the data on (data not shown) need to be included in the complementary materials.

PLOS authors have the option to publish the peer review history of their article (what does this mean? ). If published, this will include your full peer review and any attached files.

**Do you want your identity to be public for this peer review?** For information about this choice, including consent withdrawal, please see our Privacy Policy .

Reviewer #1: No

Reviewer #2: No

Reviewer #3: No

**Figure resubmission:**
---

## [Decision Letter · Decision Letter 1]

13 May 2025

Dear Dr Lapidot,

We are pleased to inform you that your manuscript entitled "Contribution of the tobamovirus resistance gene Tm-1 to control of tomato brown rugose fruit virus (ToBRFV) resistance in tomato" has been editorially accepted for publication in PLOS Genetics. Congratulations!

Yours sincerely,

Ortrun Mittelsten Scheid

Academic Editor

PLOS Genetics

Aimée Dudley

Editor-in-Chief

PLOS Genetics

Aimée Dudley

Editor-in-Chief

PLOS Genetics

Anne Goriely

Editor-in-Chief

PLOS Genetics

Comments from the reviewers (if applicable):

Reviewer's Responses to Questions

**Comments to the Authors:**

Reviewer #1: Authors have applied my recommendations to the revised ms and addressed my concerns. No further comments.

Reviewer #2: In my opinion, the authors have thoroughly addressed the comments and suggestions raised during the review process, including those from the more critical reviewer. The revised version reflects a substantial improvement, both in clarity and in the scientific rigor of the work. The authors have made a commendable effort to incorporate all relevant feedback.

Reviewer #3: none

**Have all data underlying the figures and results presented in the manuscript been provided?**

Reviewer #1: Yes

Reviewer #2: Yes

Reviewer #3: Yes

PLOS authors have the option to publish the peer review history of their article (what does this mean? ). If published, this will include your full peer review and any attached files.

**Do you want your identity to be public for this peer review?** For information about this choice, including consent withdrawal, please see our Privacy Policy .

Reviewer #1: No

Reviewer #2: No

Reviewer #3: No

**Data Deposition**

http://datadryad.org/submit?journalID=pgenetics&manu=PGENETICS-D-25-00121R1

**Press Queries**

---

## [Editor Report · Acceptance letter]

PGENETICS-D-25-00121R1

Contribution of the tobamovirus resistance gene Tm-1 to control of tomato brown rugose fruit virus (ToBRFV) resistance in tomato

Dear Dr Lapidot,

We are pleased to inform you that your manuscript entitled "Contribution of the tobamovirus resistance gene Tm-1 to control of tomato brown rugose fruit virus (ToBRFV) resistance in tomato" has been formally accepted for publication in PLOS Genetics! Your manuscript is now with our production department and you will be notified of the publication date in due course.

With kind regards,

Zsofia Freund

PLOS Genetics

On behalf of:
